# Economic policy uncertainty and stock market in G7 Countries: A panel threshold effect perspective

**Maysoon Khojah[1], Masood Ahmed[2], Muhammad Asif Khan [3]\*, Hossam Haddad[4], Nidal Mahmoud Al-Ramahi[5], Mohammed Arshad Khan[6]**

1 Accounting Department, College of Administrative and Financial Sciences, Saudi Electronic University, Riyadh, Saudi Arabia, 2 Department of Public Administration, University of Kotli, Azad Jammu and Kashmir, Pakistan, 3 Department of Commerce, University of Kotli, Azad Jammu and Kashmir, Pakistan, 4 Business Faculty, Zarqa University, Zarqa, Jordan, 5 Accounting Department, Zarqa University, Zarqa, Jordan, 6 Department of Accountancy, College of Administrative and Financial Sciences, Saudi Electronic University, Riyadh, Saudi Arabia

\* khanasif82@uokajk.edu.pk

**Data Availability Statement:** The data underlying the results presented in the study are available from: Economic policy official website: http://www.policyuncertainty.com/ Yahoo Finance official

## Abstract

Based on the literature, it is commonly understood that stock prices (SP) are influenced by economic policy uncertainty (PU), with a rise in PU typically having a negative impact on SP. However, the relationship between PU and SP may not always be linear due to the varying risk preferences of individuals. Risk preference theory posits that individuals respond differently to different levels of risk. Therefore, this study aims to investigate whether PU determines SP asymmetrically (i.e., in a non-linear manner) by considering risk preferences and addressing a gap in the literature. To answer this question, the study employs a panel threshold approach to examine the effect of PU on SP in the Group of Seven (G7) countries, namely Canada, France, Germany, Italy, Japan, UK, and the US. In contrast to previous research, this study finds evidence of an asymmetric effect of PU on SP in the G7 countries. Specifically, the panel threshold results reveal that the impact of increased PU on SP is positive up to a certain level (Threshold1), beyond which it becomes negative (Threshold2). These findings are in line with information asymmetry hypothesis, prospect theory, behavioural finance hypothesis, and market liquidity hypothesis and shed light on the asymmetric behaviour of SP in response to varying levels of PU. The implications of these findings are significant for understanding how to manage risks effectively in the financial markets.

## 1. Introduction

The study is motivated by the conflicting and inconclusive empirical literature on economic policy uncertainty (PU) and stock prices (SP) dynamics, and unaddressed phenomena of risk preference theory. For example, classical literature [1] documents that, "risk taking is any consciously, or no-consciously controller behavior with a perceived uncertainty about its outcome, and/or about its possible benefits or costs for the physical, economic or psycho-social well-being of oneself or others." The risk taking behavior as a decision problem and specifies it

website: http://finance.yahoo.com/ FRED: https://fred.stlouisfed.org/.

**Funding:** This work was not supported by any agency; however, the APC shall be covered by Zarqa University, Jordan. The funders had no role in study design, data collection and analysis, decision to publish, or preparation of the manuscript.

**Competing interests:** The authors have declared that no competing interests exist.

as "a loss; the significance of loss; or the uncertainty associated with loss" [2]. Thus, behavior of stock market with response to changes in PU may not necessarily be linear, as market may respond differently to different level of risk. Theoretically, the asymmetric response may be induced by the investors risk preference, hence, the possibility of stock market asymmetric response to PU calls for testing this hypothesis empirically. For this purpose, this study uses fixed effect panel threshold modeling approach [3] to investigate the threshold effect of PU on stock market returns of Group of Seven (G7) countries.

The G7 countries, including Canada, France, Germany, Italy, Japan, the United Kingdom, and the United States, are major global economies with substantial financial markets. These countries play a significant role in shaping global economic policies and have a substantial impact on international financial markets. Findings derived from studying the G7 countries have broader implications due to their global economic significance. As these countries are influential players in the international financial system, their financial market dynamics and policy implications can often be generalized to other countries or regions. Consequently, empirical research conducted on G7 countries can provide insights that are relevant and applicable to a wider range of economies and financial markets. Therefore, the use of the G7 countries as a sample in empirical research within the finance field is driven by their economic significance, diverse policy frameworks, availability of data, generalizability of findings, benchmarking value, and policy relevance.

Since the introduction of the measure for PU [4], scholars, businessmen, and policymakers closely track PU and its impacts on the economy and stock markets [5]. PU is defined as "a non-zero probability of changes in existing economic policies that determines the rules of the game for economic agents." PU has many implications for the economy. Companies and other important economic actors change and delay their decisions regarding investment, employment, consumption, and saving due to PU [6–8]. Existing literature suggests that there is a generally accepted understanding that stock prices (SP) are influenced by changes in PU. It is widely observed that when PU increases, there tends to be a negative impact on SP. PU is expected to negatively impact stock markets because investors always look for certainty and stability. PU also has an adverse impact on financing and production costs because it affects supply and demand, leading to investment decline and economic contraction [7, 9–11]. PU increase financial risk by decreasing government protection for markets [9]. Inflation, interest rates, and risk premium are also impacted by PU [12, 13]. However, it is important to note that the relationship between PU and SP may not always follow a simple linear pattern. This can be attributed to the diverse risk preferences exhibited by individuals, as highlighted by risk preference theory.

According to the risk preference theory, investors behave differently to different level of risk. The degree of preference that an individual has for the outcomes, as well as his outlook on the value of taking risks, both have a role in determining the likelihood that the individual will choose hazardous options [14]. Gitman and Zutter [15] described the investors risk preferences, explain that different people react differently to risk, and classify investor reactions to risk into three categories. The first category, and the one that most people fall into most of the time, is known as risk aversion, that prefers lower-risk investments over higher-risk investments, while keeping the rate of return constant. A risk-averse investor who believes that two different investments have the same expected return will select the investment with the higher certainty of returns [15]. To put it another way, a risk-averse investor will not choose the riskier investment unless it offers a higher expected return to compensate the investor for taking on the additional risk. Risk neutrality is a second risk-taking attitude. A risk-averse investor chooses investments based solely on their expected returns, ignoring the risks. When deciding between two investments, a risk-averse investor will always choose the one with the higher

expected return, regardless of risk. Under risk-neutral conditions, the no-arbitrage principle dictates that the market price of an option must equal the present value of its expected future payoffs, discounted at the risk-free rate [16].

Despite the fact that risk premium are unobservable, the risk-neutral density calculated from observable option prices offers useful information regarding investors' expectations and risk preferences [17]. Finally, a risk-seeking investor prefers higher-risk investments and may even forego some expected return when selecting a riskier investment [15]. When market volatility is strong, it is likely that investors are still eager to take risks [16]. Although, this behavior if not common, yet how people process information and make decisions, including risky ones, depends on what kind of investor they are [18]. The findings of an international sample on risk preference indicate, on average, a risk-averse attitude toward gains and a risk-seeking attitude toward losses, with considerable cross-country variances in the degree of risk aversion [19].

This study extends the risk-return literature on non-linear lines by analyzing the panel threshold effect [3] of PU on Stock prices of G7 economies. The empirical findings of panel threshold regression show two significant thresholds in the context of PU and SP for G7 countries. Specifically, the panel threshold results indicate that the increased PU has a significant positive effect on stock prices in G7 countries up to a certain level (Threshold1 = 111.607 (equal to natural log 4.714)), after which the effect is negative (Threshold1 = 242.846 (equal to natural log 5.493)). The findings are consistent with risk preference theory and have implications for understanding the asymmetric behavior of SP to different level of PU.

The findings of this study have important policy implications for stakeholders in the financial markets, such as investors, policymakers, and regulators. Firstly, the results suggest that investors need to be mindful of the asymmetric effect of PU on SP in the G7 countries. They should consider the threshold levels identified in this study and adjust their investment strategies accordingly. For example, investors may want to increase their exposure to stocks when PU is below the identified threshold level and reduce their exposure when it exceeds the threshold level. This could help them to optimize their returns and manage risks more effectively. Secondly, policymakers and regulators need to be aware of the non-linear relationship between PU and SP in the G7 countries. They should consider this relationship when making economic policy decisions, such as changes to interest rates or fiscal policies. Policymakers and regulators should also ensure that they communicate policy changes clearly to investors and market participants to minimize uncertainty and reduce market volatility. Thirdly, the study's results highlight the importance of understanding individual risk preferences when managing risks in the financial markets. Investors' behavior in response to changes in PU may vary based on their risk preferences. Therefore, financial institutions should take a more nuanced approach to risk management, tailoring their strategies to the specific risk preferences of their clients. Overall, this study's findings emphasize the need for a more sophisticated and nuanced approach to risk management in the financial markets. By understanding the non-linear relationship between PU and SP and taking into account individual risk preferences, stakeholders can better manage risks and optimize returns.

Remaining work is arranged as follows. Section 2 outlines the brief theoretical and empirical literature review. Section 3 shows the methodological setting. Section 4 illustrates the empirical results. Discussion is made in section 5, and conclusion is drawn in section 6.

## 2. Literature review

### 2.1 Theoretical underpinning

There is no single theory that can fully explain the phenomenon of asymmetric effect of EPU on stock prices. It is likely that a combination of theories (factors), including information

asymmetry hypothesis [20], prospect theory [21], behavioral finance hypothesis [22], and market liquidity hypothesis [23], all play a role. This section briefly outlines few of the theories underpin the threshold (asymmetric) effect of EPU on G7 Stock Prices.

**2.1.1 information asymmetry hypothesis.** The information asymmetry hypothesis by Akerlof [20] can be used to explain the asymmetric effect of EPU on stock prices in G7 countries. When EPU is high, investors have less information about the future direction of the economy and the policies that will be implemented. This can lead to market volatility, as investors become more risk-averse and are less willing to buy stocks. The information asymmetry hypothesis suggests that the asymmetric effect of EPU on stock prices is due to the fact that investors have more information about the negative consequences of EPU than the positive consequences. For example, investors know that EPU can lead to lower economic growth, higher unemployment, and a decline in corporate profits. However, they may not be as aware of the positive consequences of EPU, such as increased government spending and investment. As a result, investors are more likely to sell stocks when EPU increases, leading to a decline in stock prices. However, they are less likely to buy stocks when EPU decreases, leading to a slower recovery in stock prices. The information asymmetry hypothesis can also help to explain why the asymmetric effect of EPU is more pronounced in bear markets than in bull markets. In bear markets, investors are already pessimistic about the future direction of the economy. As a result, they are more likely to sell stocks even when EPU decreases, which can lead to a sharper decline in stock prices. In bull markets, investors are more optimistic about the future direction of the economy. As a result, they are less likely to sell stocks even when EPU increases, which can lead to a slower recovery in stock prices.

**2.1.2 Prospect theory.** Prospect theory, developed by Kahneman and Tversky [21], is a behavioral economic theory that seeks to explain how individuals make decisions under risk and uncertainty. The theory provides insights into how people perceive and evaluate potential gains and losses, and how these perceptions influence their decision-making. In the context of the asymmetric impact of EPU on stock prices, prospect theory helps explain why individuals may react differently to positive and negative changes in policy uncertainty. According to prospect theory, individuals evaluate potential gains and losses relative to a reference point, typically their current wealth or the status quo. The theory posits that people are generally loss-averse, meaning the pain of potential losses is felt more strongly than the pleasure of equivalent gains. This leads to a bias where individuals are more motivated to avoid losses rather than maximize gains. Applying prospect theory to the asymmetric impact of EPU on stock prices, we can expect to observe a stronger negative reaction to increases in policy uncertainty compared to positive reactions to decreases in uncertainty. When policy uncertainty rises, it creates an environment of increased risk and ambiguity. Loss-averse investors, concerned about potential downside risks, may become more risk-averse and sell their holdings or reduce their exposure to the stock market. This selling pressure can lead to larger declines in stock prices.

Conversely, when policy uncertainty decreases, it can provide a sense of relief and reduce perceived risks. However, the positive impact on stock prices may not be as pronounced as the negative impact because of loss aversion. Investors may be less motivated to take on additional risk and increase their stock holdings to capitalize on the decrease in uncertainty, leading to a smaller or more muted positive effect on stock prices. To summarize, prospect theory suggests that individuals' responses to changes in policy uncertainty are asymmetric due to their loss aversion. Negative changes or increases in uncertainty are likely to result in stronger negative reactions and larger declines in stock prices, while positive changes or decreases in uncertainty may have a smaller or more muted positive impact on stock prices.

**2.1.3 The behavioral finance hypothesis.** Behavioral finance hypothesis [22] states that investors are not always rational and may make decisions based on emotions, such as fear and greed. When EPU is high, investors may become more fearful and sell stocks, even if there is no fundamental reason to do so. This can lead to a sharp decline in stock prices. However, when EPU is low, investors may become greedier and buy stocks, even if there is no fundamental reason to do so. This can lead to a slower recovery in stock prices.

**2.1.4 The market liquidity hypothesis.** The market liquidity hypothesis [23] states that EPU can lead to a decrease in market liquidity, which makes it more difficult for investors to buy and sell stocks. This can lead to a sharp decline in stock prices, as investors are less able to adjust their positions in response to changes in EPU. However, when EPU is low, market liquidity is restored, which can lead to a slower recovery in stock prices.

It is important to note that the asymmetric effect of EPU on stock prices is not always straightforward. The effect may depend on a number of factors, including the specific context (sample, data, variable, and method). For example, one study may find that the asymmetric effect of EPU on stock prices is more pronounced in bear markets than in bull markets. Another study may report that the asymmetric effect is more pronounced for stocks of companies with high levels of debt. Nevertheless, the theories of information asymmetry, prospect (preference) behavioral finance, and market liquidity are still some of the most commonly referred theories in support of explaining the asymmetric effect of EPU on stock prices.

## 2.2 Empirical literature

The impact of PU on various forms of stock markets proxies has widely been studied empirically. For example, Brogaard and Detzel [24] uses the monthly value-weighted CRSP index as a measure of US stock market performance and estimates its relationship with PU. The study finds that stock returns are decreased by 1.31% with one standard deviation increase in PU. They find no relationship between dividend growth and PU. Arouri and Estay [7] finds a weak but persistent negative impact of PU on the US stock market returns. The findings of the study suggest that during high volatility, the negative impact of PU on stock returns becomes stronger. Bahmani-Oskooee and Saha [25] finds that PU impacts stock prices negatively only in the short run in thirteen countries, including the US. Kundu and Paul [26] found that PU has a larger negative impact on stock returns and a smaller positive impact on volatility in the bearish regime than in the bullish regime. This implies that PU reduces stock returns more and increases volatility less when the market is down than when it is up. A study Wen, Shui [27] investigates the impacts of PU on the time-varying risk–return relationship and explores whether this impact is significantly different during the 2008 Global Financial Crisis (GFC). The study employs a GARCH-M model and finds that both the national and global PU shocks have significant and negative impacts on the time-varying risk–return relationship, and that these negative impacts increase and intensify during the GFC.

Liu and Zhang [28] uses S&500 index data from January 1996 to June 2013 and finds that PU impacts stock market volatility and holds significant predictive power on stock market volatility. Asgharian and Christiansen [29] examines the relationship between the US (S&P500) and the UK (FTSE) stock markets and PU using daily stock market data and monthly PU data. The findings suggest that the US stock market volatility is impacted by the US PU only, whereas the UK stock market is impacted by PU in both the UK and the US. Wu and Liu [30] analyzes monthly data from Jan 2013 to December 2014 of Canadian, Spanish, the UK, French, Italian, Chinese, Indian, the US, and the German stock market. The findings of the study suggest that not all markets react similarly to PU. UK stock market falls the most with negative PU, but Canadian, the US, French, Chinese, and German stock markets remain mostly

unaffected. Gao and Zhu [31] considers both domestic and international PU factors and examines how this impact the UK stock market. The study argues that PU explains the cross-section of the UK stock market returns. Consistently, the major studies on PU and stock markets measures are illustrated in Table 1 with response to the country, period, variables, method, and findings. The theme of tabular presentation of core literature is adopted from Kumar, Chandra [32].

The studies conducted on the impact of PU on stock market measures have yielded mixed results (as reported in Table 1), with both negative and positive findings, and most of the literature is devoted to linear impact. These divergent outcomes suggest that the relationship between PU and stock markets is inconclusive and dependent on the specific context or conditions under investigation. In other words, the impact of PU on stock market dynamics may vary in different countries, time periods, or economic environments. The behavior of stock markets in response to changes in PU is not necessarily linear, as market reactions can vary depending on the level of risk. This asymmetric response may be influenced by investors' risk preferences, leading to the need for empirical testing of this hypothesis. In this study, a fixed effect panel threshold modeling approach is employed to examine the threshold effect of PU on SP in the G7 countries. By utilizing this methodology and sample, the study aims to fill the knowledge gap in assets pricing literature.

Fig 1 shows the average monthly SP of G7 countries for sample period, while Fig 2 depicts the average monthly PU.

Fig 3 illustrates the Theoretical Framework Drawn on Risk Preference theory and empirical literature.

## 3. Research method

According to literature [55], threshold models are visionary innovation, and advancements in the fields of economics and econometrics are greatly influenced by this model. Threshold regression can capture the threshold(s) point where the magnitude and direction of the impact of PU on stock prices changes. Most of the literature on PU and the stock market uses linear modeling, while the Threshold models are increasingly used to capture nonlinear behavior in economics. In panel dataset, researchers often used traditional fixed effect or random effect models guided by Hausman test. Panel data is plagued by the problem of heterogeneity. To put it another way, each unit in a study is unique, and the structure of their interactions may differ. Only the variability in intercepts is reflected by the standard fixed or random effect. That is the reason present study uses panel threshold regression to examine the threshold effect of PU on stock prices of G7 countries. In panel threshold regression, there is one effect (one set of coefficients) up to the threshold and another effect (another set of coefficients) beyond the threshold, as delineated by thresholds. The leaping character or structural break in the relationship between variables is characterized and described by the threshold model. As a result of their clear and unambiguous economic implications, threshold models are frequently utilized in macroeconomics and financial research.

Single threshold model is specified as under;

$$y_{it} = \mu + X_{it}(q_{it} < \gamma)\beta_1 + X_{it}(q_{it} \geq \gamma)\beta_2 + \mu_i + e_{it} \tag{1}$$

Where, $q_{it}$ is threshold variable, and $\gamma$ as threshold parameter, with coefficients of $\beta_1$ and $\beta_2$. The threshold variable, $q_{it}$, separate the equation into two regimes. Individual effects are represented by the parameter $\mu_i$, whereas disturbance term is represented by $e_{it}$.

**Table 1.**

| Reference | Country | Period | Variables | Method | Findings |
|---|---|---|---|---|---|
| Sum [33] | EU | 1985–11 | PU, stock return | VAR model with Granger-causality | PU has a negative impact on stock market returns |
| Tang and Wan [34] | China | 2005–2017 | PU, stock price | Panel regression with fixed effects and instrumental variables | The positive relationship between PU and stock price informativeness is found, and unexpected PU is important. |
| Du, Sui [35] | China | 2005–2017 | PU, stock price crash risk | Panel regression with fixed effects and two-stage least squares | The increasing of PU will significantly increase stock price crash risk. Economic policy affects stock price crash risk mainly by influencing information asymmetry. |
| Ono [36] | OECD and non-OECD countries | 1997–2019 | PU, stock indices | Lag-augmented VAR model with time-varying Granger causality test | The causal relationship between the two variables is not present for the entire period, but it is present for some sub-periods in many countries. |
| Xu, Wang [37] | China and US | 2000–2019 | PU, stock market returns | VAR model with impulse response function and variance decomposition analysis | US PU has a significant negative impact on Chinese stock market returns in the short term but not in the long term. Chinese PU has a significant negative impact on US stock market returns in both the short term and the long term. |
| Cheng, Wang [38] | China | 2009–2018 | PU exposure, stock price bubbles | Panel regression with fixed effects and instrumental variables | There exists a significantly positive relationship between PU exposure and stock price bubbles. Retail trades play an important role in the transmission mechanism of PU exposure to stock price bubbles. |
| Liao, Hou [39] | Western Balkan countries | 2006–2018 | PU, stock returns | GARCH models with dummy variables for PU shocks | PU has a negative and significant impact on stock returns and a positive and significant impact on volatility |
| Fang, Bouri [40] | Global | 2010–2016 | Bitcoin returns and volatility, global PU | GARCH models | GEPU has a positive and significant impact on Bitcoin volatility and reduces the hedging effectiveness of Bitcoin against major currencies |
| Chen, Jiang [41] | China | 2000–2014 | PU, stock market expected returns | Fama-French three-factor model and GARCH-M model | PU has a negative and significant impact on stock market expected returns and a positive and significant impact on conditional volatility |
| Aydin, Pata [42] | Brazil, Russia, India, China | 2003–2021 | PU, stock prices | Asymmetric and symmetric frequency domain causality tests | There exists a significantly positive relationship between PU exposure and stock price bubbles. |
| Chiang [43] | USA, EU, China, Japan | 1990–2018 | US categorical PU, stock returns | GED-GARCH model | US policy uncertainty changes have a negative impact on markets in the US and its spillovers to the EU, China and Japan. |
| Balcılar, Demirer [44] | GCC countries (Bahrain, Kuwait, Oman, Qatar, Saudi Arabia, UAE) | 2005–2013 | Equity sector returns, global and regional PU | VAR model with Cholesky decomposition for the impulse responses and variance decompositions | PU has a negative and significant impact on equity sector returns; the impact varies across sectors and countries; the spillovers from global and regional PU are significant but asymmetric |
| Bannigidadmath and Narayan [45] | 18 emerging markets | 1990–2014 | Stock returns, dividend yield, PU | Predictive regressions with Newey-West standard errors and Hansen-Hodrick standard errors | PU has a negative and significant impact on stock returns; the impact is stronger for countries with higher financial development and lower political risk |
| Das and Kumar [46] | 17 Emerging market and developed countries | 1998–2017 | Domestic PU, US PU, Stock Prices | Multiple and partial wavelet coherence techniques | The study finds that the impact of both domestic and US PU are less significant in emerging markets than in developed markets. The study also finds that the stock prices in Canada and Australia are less sensitive to US PU. |

(*Continued*)

**Table 1.** (Continued)

| Reference | Country | Period | Variables | Method | Findings |
|---|---|---|---|---|---|
| Yang and Jiang [47] | China | 1995–2014 | PU, Stock Prices | VAR and SVAR | The study finds that PU and Chinese stock returns are negatively correlated, and this negative impact of PU on stock returns lasts about eight months after policy is announced |
| Antonakakis, Chatziantoniou [48] | US | 1985–2013 | VIX data, S7P500 Returns, Stock Prices | Dynamic conditional correlation (DCC), univariate GARCH | The findings suggest that except for the 2008 financial crisis, the relationship between PU and the stock returns is negative. |
| Li, Balcilar [49] | India and China | 1995–2013 in China and 2003–2013 in India | PU, Stock Prices | bootstrap Granger full sample causality testing and sub-sample rolling window estimation | The study finds a weak relationship that goes bidirectional for many sub-periods. |
| Škrinjarić and Orlović [50] | Nine eastern European Countries | 2001–2019 | PU, Stock Returns | Rolling estimation of the VAR model and spillover indices | The study finds that the stock markets of Lithuania, Slovenia, and the Czech Republic are more sensitive to PU shocks while the Bulgarian stock market is least sensitive. Other markets have individual reactions, according to the study. |
| Ehrmann and Fratzscher [51] | Multi-Country | 1994–2004 | US monetary policy and stock return | OLS, and PCSE | The study finds that there is a weak association between US monetary policy shocks and stock market returns of India, China, and Malaysia and strong between US monetary policy shocks and stock market returns of Korea, Turkey, Hong Kong, Indonesia, Sweden, Finland, Canada, and Australia. |
| Debata and Mahakud [52] | India | 2003–2016 | PU, stock market liquidity | VAR, Granger causality tests, impulse response functions and variance decomposition analysis | The empirical findings suggest that economic policy uncertainty moderately influences stock market liquidity during normal market conditions. However, the role of economic policy uncertainty for determining stock market liquidity is significant in times of financial crises. |
| Christou, Cunado [53] | Australia, Canada, China, Japan, Korea and the US | 1998–2014 | PU, Stock Returns | Stochastic search specification selection, Bayesian panel VAR model | The study finds that PU impacts the stock market returns negatively. The study also finds that PU in the US also impacts the stock markets of other countries negatively, except Australia |
| Dakhlaoui and Aloui [54] | Brazil, Russia, India, China | 1997–2011 | US PU, BRIC Stock Indices | (i) the GARCH(1,1), (ii) the EGARCH (1,1) (exponential GARCH model) of Nelson (1991), (iii) the T-GARCH(1,1) (Threshold GARCH model), (iv) the TS-GARCH(1,1), and (v) the P-ARCH (1,1). | The research reveals that the average return spillover from the BRIC stock indices to US uncertainty is unfavorable, while the spillover of volatility fluctuates between negative and positive values. |

There is possibility of multiple thresholds for which Eq (1) may be extended as;

$$y_{it} = \mu + X_{it}(q_{it} < \gamma_1)\beta_1 + X_{it}(\gamma_1 \leq q_{it} < \gamma_2)\beta_2 + X_{it}(q_{it} \geq \gamma_2)\beta_3 + \mu_i + e_{it} \qquad (2)$$

In this case, $\gamma_1$ and $\gamma_2$ represent the thresholds that separate the equation into its three distinct regimes, which are denoted by the coefficients $\beta_1$, $\beta_2$, and $\beta_3$. The sequential estimator is consistent, as stated by Bai [56] and Bai and Perron [57], hence the thresholds are estimated as follows:

- In order to derive the threshold estimator $\gamma_1$ and the RSS $S1(\hat{\gamma}_1)$, it is necessary to fit the single-threshold model.

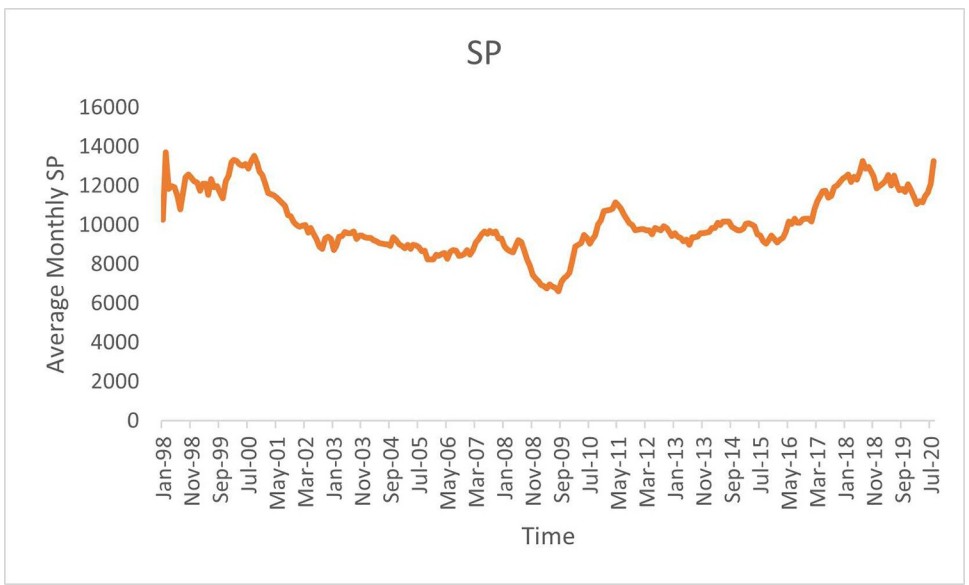

**Fig 1. Average monthly stock prices of G7 countries.**

- In the presence of $\hat{\gamma}_1$, estimating the second threshold and its confidence interval.

Similarly, if the null hypothesis is rejected for the single-threshold model, we must then test the double-threshold model. There is a single-threshold null hypothesis and a double-threshold alternative hypothesis. In order to calculate the F statistic, the bootstrapping design is quite comparable to the one used in the model with a single threshold. The procedure is essentially the same for models with more than two threshold parameters [3, 58].

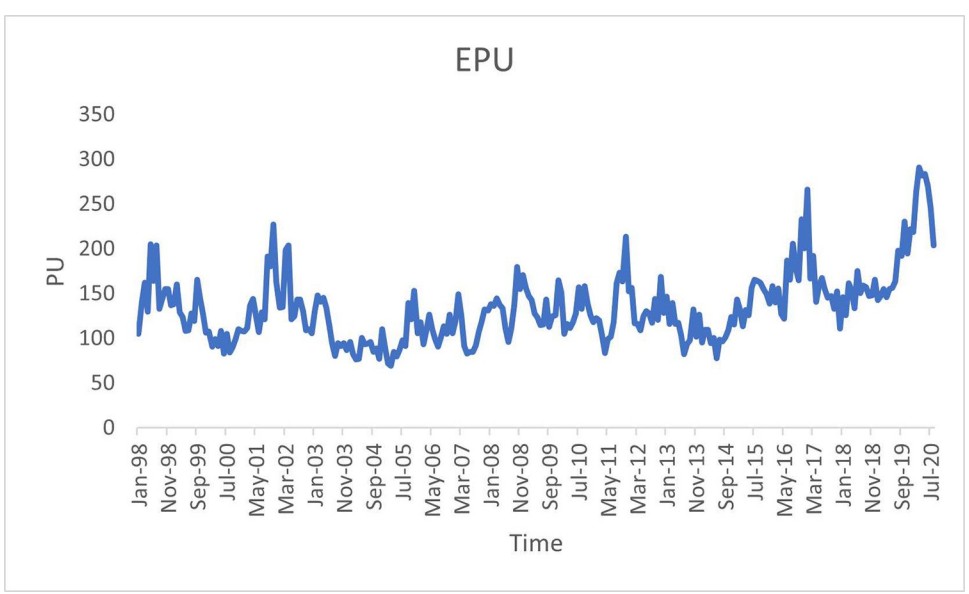

**Fig 2. Average monthly PU for G7 countries.**

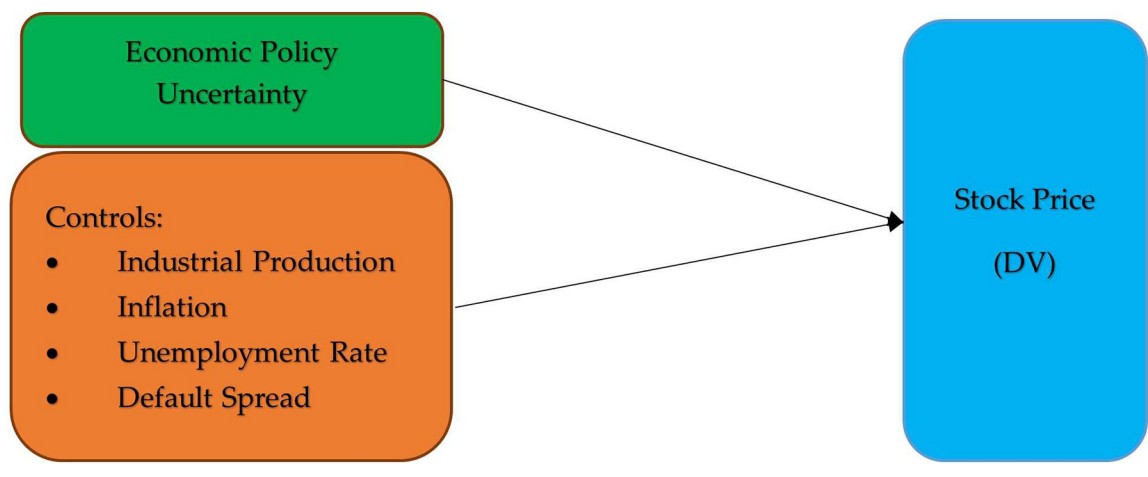

**Fig 3. Theoretical framework, IV: Independent variable, DV: Dependent variable.**

### 3.1 Variables and data source

To examine the threshold effect of PU on SP of G7 countries (Canada, France, Germany, Italy, Japan, United Kingdom, and Unites States), as shown by theoretical framework the PU is taken as an independent variable. Baker and Bloom [59] created it from using information from major newspapers of each of the nation. For example, for United States it is constituted based on three types of fundamental components. The first component assesses coverage of policy-related economic concerns in the press (by monthly searching key term "uncertainty or uncertain; economic or economy; congress, legislation, white house, regulation, federal reserve, or deficit http://www.policyuncertainty.com/us_monthly.html" from 10 US newspapers). A second component indicates the amount of elements of tax legislation that will expire in the next years. The third component uses economic meteorologists' disagreement as an indicator of uncertainty. First, each component is normalized by its own standard deviation to produce overall index of policy-related economic uncertainty. After that, an average is taken, giving the tax expirations index, the CPI prediction disagreement measure, and the federal/state/local purchases disagreement measure each a weight of 1/6 and giving the broad news-based policy uncertainty index a weight of 1/2. Similarly, the index construction relies on alike methodology with different count of number of newspapers.

This study makes use of monthly data spanning from January 1998 all the way through August 2020. For this time period, data on PU were obtained by downloading them from the PU website, which can be found at http://www.policyuncertainty.com/. This website is an open source and is frequently referenced in the relevant research, and for each of the G7 member country the respective details may be accessed at https://www.policyuncertainty.com/. The SP (stock price) of each of G7 countries are the dependent variable for present analysis. The monthly closing SP of one major stock market index of each of G7 countries is downloaded from Yahoo Finance (http://finance.yahoo.com/).

Aligned with related literature [7, 25, 60], this study included inflation (INF), industrial production (IP), unemployment rate (UE), default spread (DS), and gross domestic product (GDP) as probable controls to overcome the possible overestimation. The secondary monthly data for all control variables are downloaded from Federal Reserve Economic Data (FRED https://fred.stlouisfed.org/) [61]. FRED being an open database maintains a wide variety of datasets from around 100 data sources pertaining to more than 0.07 million worldwide

indicators. other than DS, and UE all the variables used in the study are transformed to natural log for smooth analysis.

## 4. Results and discussion

### 4.1 Descriptive statistics

Descriptive statistics are reported in the Table 2. As evident in Table 1, all the test variables have a positive average value, with a low standard deviation except Default Spread (DS) where standard deviation is reportedly higher. With regard to Unemployment Rate (UE), we observe a positive mean value and a moderately high value of standard deviation.

### 4.2 Threshold effect

We use the panel threshold model to analyze the possible asymmetric nonlinear relationship PU and SP. The threshold effect was tested first, using panel fixed effect threshold estimator of Wang [3] before estimating the threshold regression. There are two tests, F and LR, where the former examines the significance of the threshold effect, and the latter examines whether the threshold estimation value is equal to the true value. The two tests' results are presented in Table 2. The p-values of TH1 and TH2 are significant with their individual threshold values are 4.714 and 5.493 respectively, thus confirming the presence of two significant threshold in the estimation of effect of PU on SP in G7 countries. Two significant thresholds are graphically shown in Fig 4, where blue dots are showing overall pattern of movement in PU, while two circles on timeline are designated by red arrowhead. The circles on timeline are significant thresholds where the relationship of PU and SP in context of G7 countries turns non-linear. At the tail of arrows respective significant threshold values are reported, which are the same as in Table 3.

Table 4 presents the estimation output for our fixed effect regression model. The estimations control for various control variables namely INF, IP, DS, UE and GDP. The main variable of interest is the PU, that is found to have a significant and negative impact on stock prices. To better illustrate the non-linear effect of the explanatory variable, PU we estimated fixed effect model in three scenarios. First estimated model reports the results of impact of PU below first threshold (TH1) witnessing that until a certain level, the PU positively explains the SP with a coefficient of 0.492. The results of first model confirm that a certain level of PU (below first threshold) encourages the trading of given stock market instruments. The results of fixed effect estimation between first and second threshold (PU>TH1<TH2) are reported in column (2) providing evidence of adverse implication of rising PU on SP where coefficient turns negative.

The negative and significant coefficient of -0.221, signals that stock markets don not like PU above a certain level (above first threshold, and below second threshold). More precisely,

| Country | Stock market index |
|---|---|
| 1. Canada | S&P/TSX Composite Index (GSPTSE) |
| 2. France | CAC 40 (FCHI) |
| 3. Germany | DAX PERFORMANCE-INDEX (GDAXI) |
| 4. Italy | FTSE MIB Index (FTSEMIB.MI) |
| 5. Japan | Nikkei 225 (N225) |
| 6. United Kingdom | FTSE100 |
| 7. United States | Dow Jones Industrial Average (DJI) |

**Table 2. Descriptive statistics (actual data).**

| Variable | Mean | Std.Dev. | Min | Max |
|---|---|---|---|---|
| SP | 12080.67 | 8311.5 | 2423.87 | 48479 |
| PU | 135.462 | 78.679 | 11.287 | 678.817 |
| INF | 109.191 | 44.36 | 70.4 | 259.918 |
| IP | 101.523 | 10.33 | 5.56 | 133.389 |
| DS | 15.512 | 34.635 | -1.233 | 106.127 |
| UE | 6.919 | 2.419 | 2.2 | 14.7 |
| GDP | 99.963 | 1.711 | 74.833 | 103.18 |
| Descriptive Statistics (Natural Log) | | | | |
| SP | 9.194 | 0.635 | 7.793 | 10.789 |
| PU | 4.766 | 0.532 | 2.424 | 6.520 |
| INF | 4.635 | 0.312 | 4.254 | 5.560 |
| IP | 4.614 | 0.121 | 1.716 | 4.893 |
| DS* | 15.512 | 34.635 | -1.233 | 106.127 |
| UE* | 6.919 | 2.419 | 2.200 | 14.700 |
| GDP | 4.605 | 0.018 | 4.315 | 4.636 |

*Note*: SP: Stock Price; PU: Economic Policy Uncertainty; INF: Inflation; IP: Industrial Production; DS: Default Spread; and GDP: Gross Domestic Product

*indicates that these variables are not transformed to natural log

Source: Based on authors' estimation

the case where PU is above first threshold (PU>TH1), the coefficient is highly negative and significant. This implies that the demand for stock market securities is threatened by increase in PU that pushes the risk averse investor to sell their securities that on the other hand increases the supply causing downward movement in equilibrium prices. The finding are consistent with the law of demand and also gains support from empirical literature as predicted by

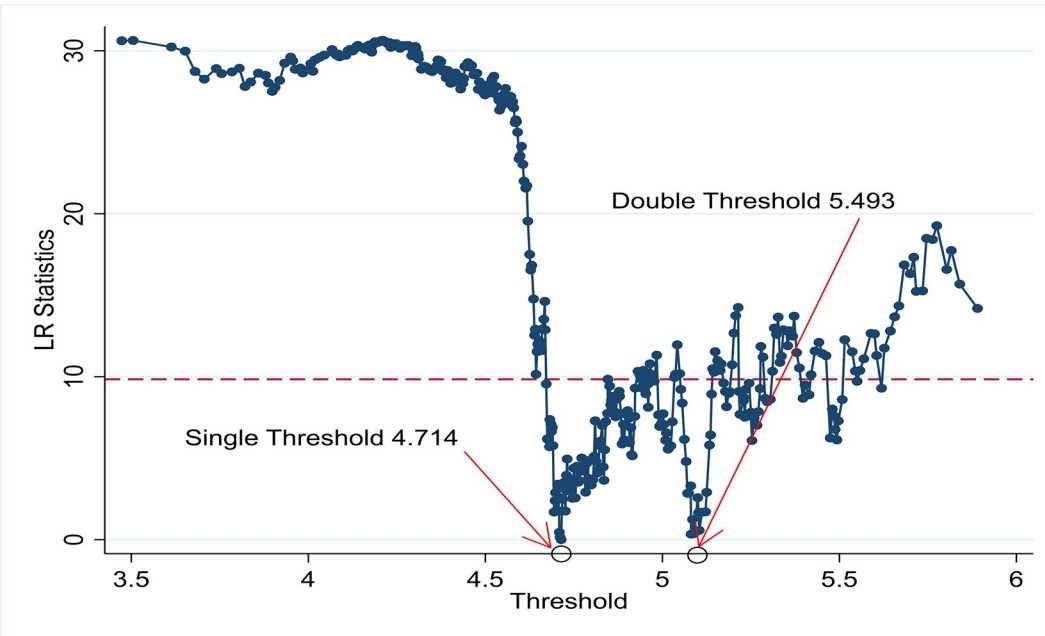

**Fig 4. Threshold and LR statistics.**

**Table 3. Panel threshold regression.**

| Model | Threshold | Lower | Upper | F-stat | P-value | Crit10 | Crit5 | Crit1 |
|-------|-----------|-------|-------|--------|---------|--------|-------|-------|
| Th-1 | 4.7140 | 4.6965 | 4.7160 | 27.14 | 0.000 | 20.671 | 22.2930 | 25.765 |
| Th-2 | 5.4930 | 5.4650 | 5.4970 | 15.70 | 0.020 | 11.329 | 14.167 | 18.804 |

Ko and Lee [62] and Pastor and Veronesi [9] who report a similar negative relation between PU and SP. The negative effect may be interpreted as that government policies affect many firms, so the effect of government policy as a result of rising PU is a systematic risk that cannot be diversified away. In Model (3) where PU exceeds TH2, the coefficient becomes positive but insignificant. The situation shows that during the period between TH1 to TH2, majority of the pessimistic investors might have shifted the investment to bond market where the return has increased due to increase in PU and remain risk seeking investors are involved in trading but due to low volume of trading the stock markets are not gaining considerable momentum. The findings of threshold analysis are aligned with the intuition and philosophy of risk return theorem, and effect of risk aptitude of investors.

**Table 4. Panel fixed effect regression results.**

| | (1) | (2) | (3) |
|---|---|---|---|
| **Variables** | SP | SP | SP |
| PU<TH1 | 0.492*** | | |
| | (0.073) | | |
| PU>TH1<TH2 | | -0.221*** | |
| | | (0.070) | |
| PU>TH2 | | | 0.099 |
| | | | (0.097) |
| INF | 0.030 | 0.378*** | 0.783*** |
| | (0.085) | (0.045) | (0.095) |
| IP | 3.581*** | 3.203*** | 0.050 |
| | (0.179) | (0.156) | (0.152) |
| DS | -0.009*** | -0.007*** | -0.005*** |
| | (0.001) | (0.000) | (0.001) |
| UE | -0.052*** | -0.017*** | -0.116*** |
| | (0.010) | (0.006) | (0.011) |
| GDP | 0.706 | -3.853*** | -0.839 |
| | (2.045) | (1.233) | (0.803) |
| Constant | -12.326 | 11.765** | 9.346** |
| | (9.296) | (5.524) | (3.813) |
| Observations | 520 | 1003 | 381 |
| R-squared | 0.597 | 0.455 | 0.376 |
| F-statistics | 64.144 | 232.735 | 78.404 |
| Cross-sections | 7 | 7 | 7 |

*Note*: Standard errors are in parenthesis.

*** $p<0.01$

** $p<0.05$

* $p<0.1$

**Table 5. Robustness: Alternative estimator Driscoll-Kraay and panel corrected standard errors.**

| | (1) | (2) | (3) | (4) | (5) | (6) |
|---|---|---|---|---|---|---|
| Estimator | Driscoll-Kraay standard errors | | | Panel corrected standard errors | | |
| Variables | SP | SP | SP | SP | SP | SP |
| PU<TH1 | 0.492*** | | | 0.492*** | | |
| | (0.103) | | | (0.101) | | |
| PU>TH1<TH2 | | -0.221** | | | -0.221*** | |
| | | (0.090) | | | (0.075) | |
| PU>TH2 | | | 0.099 | | | 0.099 |
| | | | (0.187) | | | (0.109) |
| INF | 0.030 | 0.378*** | 0.783*** | 0.030 | 0.378*** | 0.783*** |
| | (0.096) | (0.039) | (0.149) | (0.045) | (0.021) | (0.042) |
| IP | 3.581*** | 3.203*** | 0.050 | 3.581*** | 3.203*** | 0.050 |
| | (0.183) | (0.129) | (0.244) | (0.119) | (0.118) | (0.073) |
| DS | -0.009*** | -0.007*** | -0.005*** | -0.009*** | -0.007*** | -0.005*** |
| | (0.001) | (0.000) | (0.001) | (0.000) | (0.000) | (0.000) |
| UE | -0.052*** | -0.017** | -0.116*** | -0.052*** | -0.017*** | -0.116*** |
| | (0.017) | (0.008) | (0.021) | (0.009) | (0.006) | (0.010) |
| GDP | 0.706 | -3.853* | -0.839 | 0.706 | -3.853** | -0.839 |
| | (3.364) | (1.960) | (0.775) | (1.644) | (1.595) | (0.751) |
| Constant | -12.326 | 11.765 | 9.346** | -12.326 | 11.765 | 9.346** |
| | (15.191) | (9.104) | (4.486) | (7.509) | (7.364) | (3.678) |
| Observations | 520 | 1003 | 381 | 520 | 1003 | 381 |
| R-squared | 0.597 | 0.455 | 0.376 | 0.597 | 0.454 | 0.376 |
| F/Wald statistics | 1307.06 | 1166.97 | 377.66 | 2898.70 | 2489.11 | 857.66 |
| Cross-sections | 7 | 7 | 7 | 7 | 7 | 7 |

*Note*: Robust standard errors are in parenthesis

*** p<0.01

** p<0.05

* p<0.1

### 4.3 Robustness check

In order to examine the long-term relationship between variables, we used fixed effects with Driscoll-Kraay standard errors and panel corrected regression due to an estimated potential for errors as robustness check [63]. Table 4 represents the simulation results of panel data model with fixed effect where Driscoll-Kraay method was used to calculate error. Overall results reported in the Table 4 confirms that findings are robust in terms of the sign and significance of various threshold of PU. In general, the results reported in Table 5 are consistent with those reported in the baseline regressions, suggesting that the results are not driven by potential heteroscedasticity and serial correlation.

## 5. Discussion

In this section the findings of present study are discussed in the light of related theoretical and empirical literature.

The empirical findings show that PU boosts SP up to a certain level, above which it follows the traditional path to negatively drive the SP for G7 countries. The varying attitudes that investors have toward risk are a contributory factor that plays a role in the ups and downs in

asset values as well as the momentum of the market. Monitoring the trends of investors' risk attitudes would improve academics' comprehension of financial markets, make risk management more efficient for practitioners, and strengthen policymakers' ability to keep an eye on markets.

The empirical findings of panel threshold regression show two significant thresholds in the context of PU and SP for G7 countries. Specifically, the panel threshold results indicate that the increased PU has a significant positive effect on stock prices in G7 countries up to a certain level (Threshold1 = 111.607 (equal to natural log 4.714)), after which the effect is negative (Threshold1 = 242.846 (equal to natural log 5.493)). Two significant threshold levels identified through panel threshold regression are interesting for policy matters, explicitly the positive impact of PU on SP up to first threshold.

Positive impact of PU on SP: The positive effect (up to Threshold1) implies that increase in policy uncertainty results in an expected increase in risk premium that is translated into stock prices. This situation makes risk averse investor to involve in selling their stock, while risk-seekers benefit from this opportunity with an expectation to capitalize on increasing policy uncertainty with a speculation that this rise in policy uncertainty will rise risk premium in stock prices [60].

Negative impact of PU on SP: It is implied form the negative impact that that PU score between the first and second thresholds that compels pessimistic investors to sell their securities, resulting in high supply and low demand, which leads to a decline in stock prices and vice versa for a reduction in such risk [60]. It is crucial to realize that a high level of PU appears to include the majority of investors in shifting investments to relatively safe heavens, which behave differently till the first threshold. The following pathways or any combination of them might hypothetically be used to cause stock prices to alter in response to changes in PU. Important decisions by stakeholders (policymakers, regulators, enterprises, and economic agents) (e.g., employment, investment, consumption, and savings) may be delayed as a result [8, 60]. By altering supply and demand channels, it raises finance and production costs, accelerating the drop in investments and economic contraction [9, 11, 64, 65]. Finally, it is argued that the expected risk premium associated with policy decisions should be positive on average [9], which also influences inflation, interest rates, and expected risk premiums [12, 13]. The financial risk may be increased as a result of such changes. As a result, businesses that are dealing with more uncertain economic policy will make less long- and short-term investments [6].

In the same vein, the literature reports few scenarios where PU is not explaining SP [25, 30, 66]. The conflicting PU-stock market conundrum will continue to be an interesting debated in the future [60]. The findings are consistent with risk preference theory and have implications for understanding the asymmetric behavior of SP to different level of PU. By comprehending the initial PU scores of 111.607 for the single threshold and 242.846 for the double threshold, respectively, readers may carefully interpret the results of the thresholds considering the equivalent natural log transformation of 4.714, and 5.493 respectively.

Theoretically the asymmetric effect of PU and stock prices is based on the idea that PU affects the expectations and behavior of investors, firms, and policymakers in different ways depending on the direction and magnitude of the uncertainty shocks [26]. For example, a positive PU shock (an increase in uncertainty) may reduce the investors' confidence and risk appetite, leading to a decline in stock prices. However, a negative PU shock (a decrease in uncertainty) may not have the same effect, as investors may not react as strongly to good news as to bad news. This asymmetry may also depend on the market conditions, such as the level of volatility, liquidity, and sentiment. Similarly, PU may affect the investment and production decisions of firms differently depending on whether they face positive or negative uncertainty shocks [67]. Firms may postpone or cancel their investment plans when facing high

uncertainty, but they may not increase their investment when facing low uncertainty. Moreover, PU may influence the monetary and fiscal policies of the government differently depending on the state of the economy and the nature of the uncertainty [38]. For instance, the central bank may adopt a more accommodative monetary policy when facing high uncertainty to stimulate the economy, but it may not tighten its policy when facing low uncertainty to avoid overheating.

## 6. Conclusion

This study examines the impact of economic policy uncertainty on the stock prices of G7 countries (Canada, France, Germany, Italy, Japan, UK, and the US) using monthly data from January 1998 to August 2020. In order to account for the possible asymmetric effect of policy uncertainty on stock prices, the study employs panel threshold model. The empirical results reveal an asymmetric effect of policy uncertainty on stock prices in the G7 countries. Specifically, policy uncertainty has a positive impact on stock prices up to a certain level (Threshold1), beyond which the effect turns negative up to another certain level (Threshold2). The findings above Threshold1 are consistent with the general expectation that the stock market dislikes rising risk. However, the positive effect of policy uncertainty on stock prices is an interesting addition to the financial market literature. The results are consistent with information asymmetry hypothesis, prospect theory, behavioral finance hypothesis, and market liquidity hypothesis and have implications for understanding the asymmetric behavior of stock prices in response to different levels of policy uncertainty. Understanding the risk preferences of investors, whether individual or institutional, is crucial in financial market decision-making. The study's findings demonstrate the nonlinear response of stock prices to changes in policy uncertainty in G7 countries and underscore the importance of considering risk preferences when managing investments. Future research could expand the scope of analysis beyond the G7 countries to explore whether the observed effect of policy uncertainty on stock prices holds true in other countries as well. Comparing emerging and developed markets could also provide valuable insights for policymakers and investors. This would enable a more comprehensive understanding of the relationship between policy uncertainty and stock prices across a range of economic contexts and regions, and could provide important policy inputs for managing investments in different parts of the world.

## Author Contributions

**Conceptualization:** Masood Ahmed, Muhammad Asif Khan, Mohammed Arshad Khan.

**Data curation:** Maysoon Khojah, Muhammad Asif Khan, Mohammed Arshad Khan.

**Formal analysis:** Muhammad Asif Khan.

**Funding acquisition:** Hossam Haddad, Nidal Mahmoud Al-Ramahi.

**Investigation:** Maysoon Khojah, Hossam Haddad, Nidal Mahmoud Al-Ramahi.

**Methodology:** Masood Ahmed.

**Project administration:** Nidal Mahmoud Al-Ramahi.

**Resources:** Hossam Haddad, Nidal Mahmoud Al-Ramahi.

**Software:** Mohammed Arshad Khan.

**Supervision:** Masood Ahmed, Muhammad Asif Khan, Nidal Mahmoud Al-Ramahi.

**Validation:** Hossam Haddad, Mohammed Arshad Khan.

**Visualization:** Maysoon Khojah, Mohammed Arshad Khan.

**Writing – original draft:** Masood Ahmed, Muhammad Asif Khan, Mohammed Arshad Khan.

**Writing – review & editing:** Maysoon Khojah, Hossam Haddad, Nidal Mahmoud Al-Ramahi, Mohammed Arshad Khan.

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
