## [Decision Letter · Decision Letter 0]

12 Jun 2023

PONE-D-23-11753Economic Policy Uncertainty and Stock Market in G7 Countries: A Panel Threshold Effect PerspectivePLOS ONE

Dear Dr. Khan,

Thank you for submitting your manuscript to PLOS ONE. After careful consideration, we feel that it has merit but does not fully meet PLOS ONE’s publication criteria as it currently stands. Therefore, we invite you to submit a revised version of the manuscript that addresses the points raised during the review process.

We look forward to receiving your revised manuscript.

Kind regards,

Nikeel Nishkar Kumar

Academic Editor

PLOS ONE

“The funders had no role in study design, data collection and analysis, decision to publish, or preparation of the manuscript”

Reviewers' comments:

Reviewer's Responses to Questions

**Comments to the Author**

1. Is the manuscript technically sound, and do the data support the conclusions?

Reviewer #1: Yes

Reviewer #2: Partly

2. Has the statistical analysis been performed appropriately and rigorously? 

Reviewer #1: I Don't Know

Reviewer #2: Yes

3. Have the authors made all data underlying the findings in their manuscript fully available?

Reviewer #1: Yes

Reviewer #2: Yes

4. Is the manuscript presented in an intelligible fashion and written in standard English?

Reviewer #1: Yes

Reviewer #2: Yes

5. Review Comments to the Author

Reviewer #1: Line 48: Reiger and Wang

Line 54: Check reference criteria for direct quotations.

Clearly stipulate the Gap and contribution of the study.

S&P abbreviation?

Line 172 and onwards: Reference font is not consistent with the rest

Theoretical framework diagram can use different colour combinations for better clarity

Footnote 1: font is not consistent with the rest

Line 303 and 308: Avoid using bold in the text unless permitted by the journal format criteria

Line 350: Reference (and font) to be alligned with journal format requirement.

Reviewer #2: I would like to thank the authors for compiling this article entitled; "Economic Policy Uncertainty and Stock Market in G7 Countries: A Panel Threshold Effect". At presently there are some elements in the paper. Please find the comments valued towards reshaping the paper. As of currently this research needs to be relooked. My comments towards the paper are as follows:

1.In the abstract the aim of the study is highlighted but missing in the introduction part.

2.There is missing link and justification of why it was important to carry out a study based on G7 countries.

3. The first part of the introduction is aligned with the theoretical aspect of the study. I strongly suggest to move the theoretical underpinning shifted after the motivation of the paper.

4.It is important to first highlight the motivation of the paper and its distinctiveness .

5. Justify your claims of the findings with similar studies.

6.Literature review needs to be rearranged.

perhaps the authors might include a table for country specific studies and discuss the findings. in the table use heading such as Reference ,Country, Period, Variables, Method(s) and Findings. Use this study https://doi.org/10.1080/10941665.2020.1862884 a means to improve the literature and reference the table.

7. The literature seems to be basic. I would suggest to add themes in the literature. themes that are related to the study.

8. There are different types of fonts used in the study. I suggest that it should be consistent.

I wish the authors all the best in doing the revision of the paper.

6. PLOS authors have the option to publish the peer review history of their article (what does this mean?). If published, this will include your full peer review and any attached files.

Reviewer #1: No

Reviewer #2: No

---

## [Author Response · Author response to Decision Letter 0]

19 Jun 2023

The responses are attached as dedicated file "Response to reviewers."

---

## [Decision Letter · Decision Letter 1]

30 Jun 2023

PONE-D-23-11753R1Economic Policy Uncertainty and Stock Market in G7 Countries: A Panel Threshold Effect PerspectivePLOS ONE

Dear Dr. Khan,

Thank you for submitting your manuscript to PLOS ONE. After careful consideration, we feel that it has merit but does not fully meet PLOS ONE’s publication criteria as it currently stands. Therefore, we invite you to submit a revised version of the manuscript that addresses the points raised during the review process.

We look forward to receiving your revised manuscript.

Kind regards,

Nikeel Nishkar Kumar

Academic Editor

PLOS ONE

Journal Requirements:

Reviewers' comments:

Reviewer's Responses to Questions

**Comments to the Author**

1. If the authors have adequately addressed your comments raised in a previous round of review and you feel that this manuscript is now acceptable for publication, you may indicate that here to bypass the “Comments to the Author” section, enter your conflict of interest statement in the “Confidential to Editor” section, and submit your "Accept" recommendation.

Reviewer #1: All comments have been addressed

Reviewer #2: All comments have been addressed

2. Is the manuscript technically sound, and do the data support the conclusions?

Reviewer #1: Yes

Reviewer #2: Yes

3. Has the statistical analysis been performed appropriately and rigorously? 

Reviewer #1: I Don't Know

Reviewer #2: Yes

4. Have the authors made all data underlying the findings in their manuscript fully available?

Reviewer #1: No

Reviewer #2: Yes

5. Is the manuscript presented in an intelligible fashion and written in standard English?

Reviewer #1: Yes

Reviewer #2: Yes

6. Review Comments to the Author

Reviewer #1: All comments and suggestions have been addressed. Please ensure that article meets all other journal formatting requiremnts. All the best with the article.

Reviewer #2: I would like to thank the authors for providing valuable feedback towards the comments. This paper looks in shape from the previous version. I would still like point out some issues with the paper that still needs to be addressed.

Firstly my comment on adding themes in the literature. The authors can add separate section in the literature review and discuss the relevance of certain theories used in the study.

I also like to thank the authors in inserting a table in the literature. This has certainly added significance towards the paper.

The authors can also reference the article where the table is adopted from.

I wish the authors all the best in revising the paper.

7. PLOS authors have the option to publish the peer review history of their article (what does this mean?). If published, this will include your full peer review and any attached files.

Reviewer #1: No

Reviewer #2: No

---

## [Author Response · Author response to Decision Letter 1]

4 Jul 2023

Dear Editor

We have addressed the comments of reviewers 2 which are highlighted in green in revised manuscript, and brief reference is made in rebuttal letter. Reviewer 1 has not suggested any further changes. 

Response to reviewer 2:

Response:

We extend regards to anonymous reviewer for encouraging remarks and accepting our response to the comments. We address your comments which may be seen in green heighted text in revised manuscript, and brief reference is made below:

Following sentence is added on Page no 4, L#139-140:

“The theme of tabular presentation of core literature is adopted from Kumar, Chandra [1].”

Relevant theories to this study are briefly discussed in section created in literature section “2.1 Theoretical Underpinning” as per your suggestion. Please see pp 4-5, L 114-174.

References

1. Kumar, N.N., R.A. Chandra, and A. Patel, Mixed frequency evidence of the tourism growth relationship in small Island developing states: a case study of Tonga. Asia Pacific Journal of Tourism Research, 2021. 26(3): p. 294-307.

---

## [Editor Report · Decision Letter 2]

6 Jul 2023

Economic Policy Uncertainty and Stock Market in G7 Countries: A Panel Threshold Effect Perspective

PONE-D-23-11753R2

Dear Author,

We’re pleased to inform you that your manuscript has been judged scientifically suitable for publication and will be formally accepted for publication once it meets all outstanding technical requirements.

Kind regards,

Nikeel Nishkar Kumar

Academic Editor

PLOS ONE
---

## [Editor Report · Acceptance letter]

12 Jul 2023

PONE-D-23-11753R2 

Economic Policy Uncertainty and Stock Market in G7 Countries: A Panel Threshold Effect Perspective 

Dear Dr. Khan:

I'm pleased to inform you that your manuscript has been deemed suitable for publication in PLOS ONE. Congratulations! Your manuscript is now with our production department. 

Kind regards, 

on behalf of

Dr. Nikeel Nishkar Kumar 

Academic Editor

PLOS ONE